# Difficult Cases of Autoimmune Hemolytic Anemia: A Challenge for the Internal Medicine Specialist

**DOI:** 10.3390/jcm9123858

**Published:** 2020-11-27

**Authors:** Bruno Fattizzo, Juri Alessandro Giannotta, Fabio Serpenti, Wilma Barcellini

**Affiliations:** 1Hematology Unit, Fondazione IRCCS Ca’ Granda Ospedale Maggiore Policlinico, via Francesco Sforza 35, 20100 Milan, Italy; jurigiann@gmail.com (J.A.G.); fabio.serpenti@unimi.it (F.S.); wilma.barcellini@policlinico.mi.it (W.B.); 2Department of Oncology and Oncohematology, University of Milan, via Festa del Perdono 7, 20100 Milan, Italy

**Keywords:** warm autoimmune hemolytic anemia, cold agglutinin disease, intensive care unit, transplant, immunodeficiencies

## Abstract

Autoimmune hemolytic anemia (AIHA) is diagnosed in the presence of anemia, hemolysis, and direct antiglobulin test (DAT) positivity with monospecific antisera. Many confounders of anemia and hemolytic markers should be included in the initial workup (i.e., nutrients deficiencies, chronic liver or kidney diseases, infections, and cancers). Besides classical presentation, there are difficult cases that may challenge the treating physician. These include DAT negative AIHA, diagnosed after the exclusion of other causes of hemolysis, and supported by the response to steroids, and secondary cases (infections, drugs, lymphoproliferative disorders, immunodeficiencies, etc.) that should be suspected and investigated through careful anamnesis physical examination, and specific tests in selected cases. The latter include autoantibody screening in patients with signs/symptoms of systemic autoimmune diseases, immunoglobulins (Ig) levels in case of frequent infections or suspected immunodeficiency, and ultrasound/ computed tomography (CT) studies and bone marrow evaluation to exclude hematologic diseases. AIHA occurring in pregnancy is a specific situation, usually manageable with steroids and intravenous (iv) Ig, although refractory cases have been described. Finally, AIHA may complicate specific clinical settings, including intensive care unit (ICU) admission, reticulocytopenia, treatment with novel anti-cancer drugs, and transplant. These cases are often severe, more frequently DAT negative, and require multiple treatments in a short time.

## 1. Introduction

Autoimmune hemolytic anemia (AIHA) is a rare disease caused by an autoimmune attack against red blood cells. The disease is classified in warm (wAIHA, 48–70% of cases) and cold forms (cAIHA, 15–25% of cases) basing on the optimal temperature of activity of the autoantibodies and their isotypes. The remaining cases are mixed disorders [1,2,3]. The incidence of AIHA ranges from 0.8 to 3/100,000 per year and was estimated to be 0.81/100,000 (95% CI 0.76–0.92) per year in a recent French pediatric study [4]. AIHA may be secondary to a variety of conditions including systemic autoimmune diseases (i.e., 10% of cases with systemic lupus erythematosus), and lymphoproliferative syndromes, particularly chronic lymphocytic leukemia (5–10%) [5,6,7,8]. All types of AIHA can be acute and transient, or chronic with multiple relapses and therapy lines. Particularly at onset and in the acute setting, AIHA may present to the emergency room and is usually admitted to the general ward becoming a challenge for the internal medicine specialist. In this review, we will briefly describe the typical warm and cold AIHA presentation and diagnosis and then will focus on the difficult cases from both a diagnostic and therapeutic point of view.

## 2. Typical AIHA Presentation and Differential Diagnosis 

*Clinical vignette 1*: A 32-year-old female was admitted to the emergency room due palpitation, asthenia and dyspnea. The patient reported dark urine and yellowish sclerae in the previous week. Vital signs were normal, except for tachycardia; chest X-ray was unremarkable, and electrocardiogram showed unspecific repolarization alterations. Blood counts revealed hemoglobin (Hb) of 5.9 g/dL, MCV 107 fL (normal range, NR 87–97), normal leukocytes and platelets. During workup for transfusion, the direct antiglobulin test (DAT) was found positive for immunoglobulins G (IgG) and complement at low titer. Further testing showed LDH 1.5× upper limit of normal (ULN), reticulocytes 250 × 10^9^/L (NR 25–75 × 10^9^/L), and bilirubin of 3.1 mg/dL (NR 0.2–1.1 mg/dL), and normal vitamin (B12 and folate) and iron levels (iron, ferritin, transferrin, transferrin saturation 30%). The patient was transfused with symptoms improvement, admitted to the ward, and 1 mg/Kg day intravenous methylprednisolone was started.

The differential diagnosis of anemia in admitted patients encompasses a large number of conditions, and the initial work up should include a thoughtful clinical history and physical examination and several tests to investigate vitamin and/or iron deficiency, liver and kidney diseases, infections, hemoglobinopathies, cancers, drugs, autoimmune and hematologic diseases, etc. AIHA should be suspected in patients presenting with macrocytic anemia of various degree and altered hemolytic markers: increased LDH and unconjugated bilirubin, decreased haptoglobin, and augmented absolute reticulocytes. Each marker has possible confounders that should be excluded basing on clinical evaluation and laboratory tests (i.e., B12 deficiency for high LDH, liver failure and congenital hypo-haptoglobinemia for low haptoglobin, Gilbert syndromes for increased unconjugated bilirubin, and bleeding for reticulocytosis) [9]. The next step and gold standard for the diagnosis of AIHA is the Coombs test or direct antiglobulin test (DAT) with monospecific antisera (anti-IgG, anti-IgA, anti-IgM, anti-complement (anti C)) [10] that enables the classification of the disease. wAIHA is typically DAT positive for anti-IgG (optimal temperature of activity 37 °C), or IgG plus C, while cAIHAs are due to IgM (thermal range 4–37 °C), and the DAT is positive for C3d. Mixed forms show both characteristics of wAIHA and cAIHA, with a DAT positive for both IgG and C and high titer cold agglutinins. Finally, there is a heterogeneous group of atypical AIHAs (about 10%) that include DAT negative, IgA driven, and warm IgM types [9,10]. All these forms may have a variable degree of anemia, hemolysis and bone marrow compensation. Importantly, autoantibodies directed against autologous erythrocytes and detected by DAT, have to be distinguished from alloantibodies that may be found in patient serum and react with donor erythrocytes in the indirect anti-globulin test (IAT). The latter may be positive also in a fraction of AIHA patients as described later. Figure 1 depicts the clinical and laboratory findings typical of AIHA at onset that may be slightly different in warm and cold cases. In particular, wAIHA patients are usually younger than cAIHA, more frequently female, tend to be acute, almost invariably requiring therapy (clinical vignette 1). Extremely severe anemia (Hb < 6 g/dL) is more frequently observed in wAIHA, where the pathogenic IgG is active at body temperature of 37 °C, opsonizes the erythrocyte and causes extravascular hemolysis in the spleen. In the described case, notwithstanding the severe presentation and the transfusion need, it is important to define the type of AIHA as treatments are different [11,12,13]. wAIHA responds well to steroids (80%), and rituximab can be added as early second line in refractory cases (either low fixed dose of 100 mg/week for 4 weeks, or standard dose of 375 mg/sm/week for 4 weeks) or reserved for relapses (with about 70–80% responses). Splenectomy is another effective second-line tool (about 70% responses) to be considered in young low comorbid patients after vaccination against capsulated organisms [13].

*Clinical vignette 2*: An 82-year-old man was referred from a General Practitioner due to moderate macrocytic anemia (Hb 9.2 g/dL, MCV 103 fL). Further testing showed altered hemolytic markers with LDH 1.8 × ULN, Reticulocytes 120 × 10^9^/L, with slightly reduced folate levels (NR 3–17 ng/mL), and normal B12 (NR 130–800 ng/L) and iron values (iron NR 50–150 mcg/dL, transferrin NR 240 a 360 mg/dL, ferritin 20–200 ng/mL). The patient reported Raynaud phenomenon in the last 3 years, and seasonal variation of Hb levels was noted by reviewing his blood counts. DAT was positive for C3d, and the patient was supplemented with folic acid and taught to protect from low temperatures. Bone marrow evaluation showed the typical 5–10% CD5+ B-lymphocyte infiltrate and computed tomography (CT) scan was negative for organomegalies. Given peripheral symptoms and Hb < 10 g/dL, day care rituximab therapy was administered with progressive recovery.

Differently from wAIHA, up to one third of cAIHA may suffer from mild compensated anemia, with occasional drops during the cold seasons or after exposure to low temperatures, infections, or other stressors. cAIHA patients would often experience disabling symptoms due to the agglutination of erythrocytes at lower temperatures that occurs in the peripheral circulatory districts of fingers, nose, ears, etc. IgM autoantibodies also fix complement and may induce intravascular hemolysis, although extravascular hemolysis in the liver remains the main pathogenic mechanism [3,13]. Circulatory symptoms may cause the patient to firstly present to the rheumatologist and pose the differential diagnosis with Reynaud phenomenon, connective tissues diseases, and vasculitis. Screening of organ and non-organ specific autoantibodies (anti-nuclear, extractable nuclear antigens, anti-DNA, rheumatoid factor, etc.) is advised, particularly when other clinical signs (i.e., arthralgia, skin rash, photosensitivity, etc.) are observed. In addition, a lymphoproliferative syndrome should be taken into account in the differential diagnosis. Patients without disabling symptoms, Hb > 10 g/dL, and LDH < 1.5 × ULN may be followed in the clinic with vitamin supplementation and cold protection only. Those requiring treatment are candidate to rituximab standard dose, since steroids are effective only at unacceptable high doses and indicated only in the acute setting and during the diagnostic workup. Rituximab is known to induce a response in about 50–60% of cases, although mainly partial. Further lines of therapies include rituximab associations with fludarabine and bendamustine, and investigational complement inhibitors. It is worth remembering that splenectomy is ineffective in cAIHA, since hemolysis does not occur in the spleen [3,13]. 

## 3. What If DAT is Negative? 

*Clinical vignette 3*: A 55-year-old man was admitted to the general ward with pneumonia and moderate anemia (8.5 g/dL Hb levels, MCV 99 fL) with altered hemolytic markers and no nutrients deficiencies. DAT with polyspecific and monospecific anti-sera was negative. Other causes of hemolysis were excluded, and second and third levels DAT were performed being all negative (Figure 2). Liver and kidney functions were normal and reticulocyte counts adequate to Hb levels. Bone marrow evaluation showed erythroid hyperplasia and excluded a lymphoid or myeloid neoplasm. Hb decreased to 7.2 g/dL and a trial of empiric steroid therapy was started with progressive complete response. A diagnosis of DAT negative AIHA was made and the patient put on active follow up for steroid tapering. 

As shown in Figure 2, various methods of DAT are available with increasing sensitivity. Five to 10% of AIHA are DAT-negative and represent a diagnostic challenge possibly delaying proper treatment. In these cases, other causes of hemolysis should be excluded including congenital hemolytic anemias due to membrane defects (positive family history, younger age, typical peripheral blood smear with spherocytes, elliptocytes, etc.), congenital enzymopathies (diagnosed by enzymatic activity and molecular test), paroxysmal nocturnal hemoglobinuria (marked intravascular hemolysis, dark urine, abdominal pain, thrombosis, etc.), thrombotic microangiopathies (concomitant thrombocytopenia and schistocytes), mechanical (history of prosthetic valves or intravascular devices, marathon athletes, schistocytes), and toxic noxae (history of chemicals exposure). If the clinical suspicion of AIHA persists, it is recommended to ask for second-level tests in a reference center [10,14]. Of particular importance is the identification of atypical AIHAs due to warm IgM that are potent activators of complement and often detach from the red blood cell (RBC) during washing procedures, causing detrimental delay in diagnosis and therapy. Despite this rich armamentarium, a fraction of AIHA remains DAT-negative: in these cases, the diagnosis is made on the basis of a response to empiric steroid therapy, as depicted in patient 3. On the other hand, DAT may be artifactually positive in case of hypergammaglobulinemia, recent high dose immunoglobulin therapy, due to the presence of alloantibodies in recently transfused patients, in delayed hemolytic transfusion reactions, and in the hemolytic disease of the newborn [10]. Finally, the above mentioned indirect antiglobulin test is important to detect autoantibodies in patient’s serum and also to reveal the presence of alloantibodies (reported in 1/3 of AIHA and possibly causing transfusion reactions) [9,13]. The detection of a positive IAT requires further studies to identify their specificity and confirm the auto- or alloreactive nature; moreover, these studies will allow to find matched packed red cells units to either avoid transfusion reactions and further immunization.

## 4. What Secondary Settings Should Be Taken into Account?

*Clinical vignette 4:* A 55-year-old female patient was admitted due to cough and fever. Chest X-ray and serology confirmed the presence of Mycoplasma pneumonia. Blood counts showed moderate macrocytic anemia initially attributed to the septic state. The patient received antibiotics with amelioration of pneumonia. However, Hb continued to decrease (7.7 g/dL) with progressive increase of LDH. DAT was found positive for C3d and a short course of steroids was instituted with rapid response and complete recovery.

As shown in Table 1, AIHA may be secondary to a number of conditions that may trigger the production of autoantibodies and should be suspected and excluded during the initial work up. These noxae may be either exogenous, such as infections and drugs, host-related, as genetic predispositions and congenital syndromes, or multifactorial, as in the case of systemic autoimmune conditions and cancer. Concerning the first group, various infections have been associated with an increased incidence of AIHA, particularly Parvovirus B19 (associated with DAT positive hemolysis in up to 20% of cases) and hepatotropic virus, mostly HCV and possibly related to interferon therapy [15]. Moreover, cold agglutinin AIHA occurs in up to 3% of patients with infectious mononucleosis and Mycoplasma pneumoniae infection, as in the described case [13,15]. Finally, it is worth reminding paroxysmal cold hemoglobinuria, an ultra-rare form of AIHA caused by the Donath-Landsteiner biphasic hemolysin. It is almost invariably preceded by an infection, including syphilis and virus, particularly in children [13,15]. AIHA secondary to infections may have a more rapid benign course, as long as the underlying condition is properly treated. On the other hand, infections represent an important risk factor for mortality in chronic relapsing cases [12]. In addition there is a long list of drugs that have been proven or highly suspected to induce AIHA, including historical ones (α-methyldopa, procainamide, penicillins, cephalosporins, diclofenac, ibuprofen, thiazides, quinine, quinidine, metformin) and more recent molecules (cladribine, fludarabine, lenalidomide, oxaliplatin, teniposide, pentostatin) [13,15]. 

*Clinical vignette 5*: A 19-year-old boy was diagnosed with IgG+ wAIHA and treated with steroids. One year later, the patient presented with petechiae and nasal bleeding and, after excluding onco-hematologic causes and microangiopathies (thrombotic thrombocytopenic purpura, diffuse intravascular coagulation, hemolytic uremic syndrome), immune thrombocytopenia (ITP) was diagnosed. This association of ITP and AIHA is known as Evans syndrome. Steroids and intravenous immunoglobulin were administered with efficacy, but one month later, platelets dropped to 3 × 10^9^/L with bleeding. Re-evaluation of past clinical history revealed frequent upper airways infections and otitis and low levels of total Ig were noted. The patient was diagnosed with common variable immunodeficiency (CVID) and put on substitutive iv Ig treatment.

Among host-related secondary causes of AIHA, various immunodeficiencies have been identified as predisposing conditions, including CVID [16], IgA deficiency, and autoimmune lymphoproliferative syndromes (ALPS) [17]. These conditions should be suspected particularly in young patients with a history of frequent infections, mild splenomegaly and lymph nodes enlargement (more typical of ALPS), lymphopenia, and low Ig levels. In these cases, Evans syndrome is more frequent, implies a more profound immune dysregulation and often requires multiple lines of therapy. The correct diagnosis is important since the patient might be handled with iv Ig cycles, as in the cited example, and other immunosuppressive treatments may lead to severe infections and be contraindicated (as for splenectomy in ALPS). Other congenital syndromes that associate with AIHA are hemoglobinopathies (up to 6%), where immune hemolysis may cause sudden Hb drop and increased transfusion requirement. The presence of beta-thalassemia intermedia, transfusion dependence and alloimmunization has been found to be a risk factor for the development of AIHA [18]. Finally, some genetic factors have been associated with the development of anti-erythrocyte autoantibodies [19], including human leukocyte antigen (HLA) genotype [20,21], rearrangements of the variable regions of the immunoglobulin heavy and light chains [3], and cytokine polymorphisms (Table 1) [19]. 

Regarding multifactorial triggers of secondary AIHA, organ and non-organ specific autoimmune conditions are a typical association, particularly systemic lupus erythematosus, thyroid autoimmune disorders, systemic sclerosis, Sjögren syndrome, autoimmune liver disorders, and inflammatory bowel diseases [13,15]. These patients usually require therapy for the underlying autoimmune condition, such as rituximab or classic immunosuppressors (i.e., cyclosporine, mycophenolate mofetil, cyclophosphamide, and azathioprine), and should be managed together with the rheumatologist/immunologist.

*Clinical vignette 6:* A 75-year-old man was referred to the hematologist from another hospital due to relapsed wAIHA diagnosed 2 years ago and successfully treated with steroids. Blood counts showed Hb 7 g/dL, platelets 99 × 10^9^/L, and leukocytes 13 × 10^9^/L with 80% lymphocytes. Flow cytometry on peripheral blood led to the diagnosis of chronic lymphocytic leukemia (CLL). CT scan was negative for organomegalies, and the patient received standard AIHA therapy with blood counts recovery.

Lymphoproliferative disorders are a typical association of AIHA that may either precede or follow their diagnosis. CLL patients show the highest risk with up to 5–10% of cases developing AIHA [15,22], particularly in the presence of hematologic risk factors (Table 1) [19,22,23,24]. Other non-Hodgkin lymphomas (NHL) may develop AIHA, with higher frequencies in some subtypes (13–19% in angioimmunoblastic T-cell lymphoma and 50% in marginal zone lymphoma) [13,15]. As in the clinical vignette, NHL and CLL should be suspected in AIHA patients with peripheral lymphocytosis, systemic symptoms, and organomegalies, and appropriate work up (CT scan and bone marrow evaluation) should be performed. These patients are not to be confounded with primary cAIHA, where a clonal lymphoid bone marrow infiltrate is invariably present, usually <10%, without the typical mutation of lymphoplasmocytic lymphoma (MYD88) [25]. It is worth mentioning that if AIHA does not respond to first line steroids, generally lymphoma therapy is required [13,15]. 

## 5. Peculiar AIHA Presentations Deserving Specific Attention and Therapy

AIHA may complicate specific clinical settings, including ICU admission, bone marrow failure, pregnancy, treatment with novel anti-cancer drugs and post-transplant (Table 2).

### 5.1. AIHA in the Intensive Care Unit (ICU) and Hyperacute Hemolysis

*Clinical vignette 7*: A 62-year-old lady, admitted from 3 days to the internal medicine ward due to pneumonia, experienced abrupt worsening of general conditions, with atrial fibrillation, angina, dyspnea, and lethargy. Blood counts showed acute hemolysis with LDH 3 × ULN and Hb 5.7 g/dL. She was transferred to the ICU where transfusions, oxygen and fluid resuscitation were provided. Chest CT scan showed right segmental pulmonary embolism and DAT was positive for IgG and C3d. Low molecular weight heparin and steroids 2 mg/kg day were started. Hb and hemolytic parameters were checked 2 times per day without significant improvement in the next 2 days. Thus methylprednisolone 1 g boluses were administered for 3 days along with iv Ig 1 g/Kg/day in 2 days, and rituximab deferred due to concomitant infection. Hb levels stabilized, but heart, respiratory, and kidney functions worsened, and the patient died from multi-organ failure. 

AIHAs requiring ICU admission are rare cases, about 5% even in largest series. However, their management is challenging and requires close monitoring of Hb and hemolytic markers, intensive transfusion policies, and attentive collaboration among the intensive care specialist, the transfusion center and the hematologist. These patients require several therapies in a short time, including transfusions, steroid boluses, iv Ig, rituximab, erythropoietin, plasma-exchange, and urgent splenectomy [26,27], so that it is almost impossible to state which one is more successful in this “fight for life” setting. Despite all efforts, mortality is about 30–57% in recent reports [26,27]. Fatal outcome correlated with very severe anemia at onset (Hb < 6 g/dL), presence of concomitant immune thrombocytopenia, multi-treatment, acute renal failure, and infections [12]. Of course, age and comorbidities are expected to influence mortality risk, although death was not age-related in published series. The described patient also experienced a thrombotic complication that may be observed in about 15–20% of AIHAs. Thrombotic events, including severe episodes (pulmonary embolism, stroke, cardiac infarction) are generally related to active hemolysis and previous splenectomy [12,13,27]. Heparin prophylaxis should be considered in hemolytic admitted patients as far as platelets and coagulation parameters are permissive.

### 5.2. AIHA with Reticulocytopenia 

*Clinical vignette 8*: A 25-year-old boy presented with fever, without any evidence of infectious focus, severe anemia and transfusion dependence developed and diagnostic workup revealed wAIHA with reticulocyte counts of 66 × 10^9^/L. Bone marrow evaluation showed mild erythroid hyperplasia, dyserythropoiesis, without aberrant myeloid or lymphoid infiltrates. Bone marrow responsiveness index, calculated as ((absolute reticulocyte count) × (patient’ Hb/normal Hb)), was < 121 consistent with inadequate reticulocytosis. Endogenous erythropoietin (EPO) levels were also inadequate to the degree of anemia and the patient was treated with steroids and recombinant EPO epoetin alpha 40,000 UI/week and vitamin supplementation with progressive increase of reticulocytes and transfusion independence.

Bone marrow compensation of autoimmune hemolysis is being increasingly recognized as a determinant of anemia severity and as a crucial player of Hb recovery in AIHA. In fact, inadequate reticulocytosis, correlated with more severe anemia at onset and with a greater risk of relapse [12]. There are three main settings where inadequate reticulocytosis may be observed: (1) severe acute AIHA with concomitant sepsis, where bone marrow is shocked and requires more time to evoke a compensatory response, as in the clinical vignette; (2) presence of autoantibodies directed against the erythroblasts, as observed in chronic refractory AIHA cases whose bone marrow features resemble those of bone marrow failures (i.e., hypercellularity, dyserythropoiesis, increased reticulinic fibrosis) [28]; and (3) AIHA in the context of reduced bone marrow stem cell reserve, as in the elderly, in cases secondary to lymphoproliferative syndromes, and metastatic cancers. Of course, vitamin and iron deficiency may impair bone marrow responsiveness and should be excluded, as well as infections directly involving erythroid precursors, such as Parvovirus B19 and Leishmania infections. Importantly, inadequate endogenous EPO levels have been demonstrated in most AIHA cases with reticulocytopenia, and the administration of recombinant EPO induced a response in up to 70% of them. The dose is the same used in myelodysplastic syndromes (i.e., epoetin alpha or zeta 40,000 UI/week) and may be usually withdrawn once the reticulocyte crisis has been primed [28,29].

### 5.3. AIHA in Pregnancy

*Clinical vignette 9*: A 31-year-old woman presented with severe hemolytic anemia (Hb 7.9 g/dL) and DAT positive for IgG, at 8 months of pregnancy. Preeclamptic syndromes were excluded, as well as other causes. Provided fetus wellbeing, steroids 1 mg/kg day were administered with progressive amelioration of Hb levels and spontaneous delivery after 10 days of therapy. Of note, meta-steroidal diabetes and hypertension had developed and required therapy until steroids tapering and withdrawn. The newborn was healthy but displayed slightly reduced Hb levels (13 g/dL) and neonatal jaundice that required UV therapy. His DAT remained positive for about 3 weeks.

The differential diagnosis of anemia in pregnancy is particularly difficult and ranges from physiologic conditions such as hemodilution, nutrients deficiency, chronic preexisting conditions (i.e., hemoglobinopathies etc.), and very severe forms like preeclamptic syndromes and microangiopathies whose exclusion is mandatory. A complete anemia workup along with blood pressure evaluation, liver function, urine analysis, and blood smear are therefore highly recommended. The incidence of AIHA in pregnancy is unknown and few cases have been reported in the literature [30,31,32,33,34,35,36,37,38,39]. It has been estimated that 1 in 50,000 pregnancies may develop anti-erythrocyte autoantibodies with different phenotypic expression [30]. AIHA in pregnancy usually has a favorable course, responding to first line steroids and resolving after delivery [35]. Red blood cell transfusions may also be safely administered, particularly in case of fetus distress. Since IgG isotypes cross the placenta, newborns of mothers with warm AIHA may have positive direct Coombs test and hemolysis, as in the clinical vignette. However, the majority of cases resulted in delivery of healthy infants without significant hemolysis [30,31,33,34,35,36,37,38,39]. Beyond steroids, the use of iv Ig may be considered, and cyclosporine was safe and effective in a case report [32], whilst the experience with rituximab is very limited. Due to the increased risk of venous thromboembolism in patients with hemolytic anemia, prophylactic doses of anticoagulation should be considered during pregnancy and puerperium and balanced against bleeding risk [34]. 

### 5.4. AIHA after Novel Anti-Cancer Therapies

*Clinical vignette 10-part 1*: A 23-year-old boy presented to the emergency department due to severe hemolytic anemia that was found to be DAT negative. The patient had a history of relapsed Hodgkin lymphoma treated with chemotherapy, autologous transplant, anti-CD30 brentuximab and, after further relapse, with the checkpoint inhibitor nivolumab with remission (last dose 1 month before presentation). He was treated with transfusions, steroids and iv Ig without response. At day 7 rituximab at standard dose was administered and steroid dose increased to 2 mg/Kg. At day 20 no response was observed, and platelets dropped to 2 × 10^9^/L. Plasma exchange was then performed (total 5 courses) with progressive improvement of Hb, platelets, and hemolytic markers.

Along with their efficacy, novel anti-tumor therapies carry new toxicities, including autoimmune complications that may present to the internal medicine specialist. In particular, immunotherapies, such as checkpoint inhibitors (CPI, nivolumab, pembrolizumab, ipilimumab, and atezolizumab), are intended to reactivate host immune system against various cancers, including melanoma, pancreatic and thyroid cancers, and lymphomas. The re-activation of T lymphocytes may in turn provoke immune-related potentially fatal adverse events [40,41,42], with AIHA being the most common [41]. The median time to AIHA onset was 50 days and concomitant thrombocytopenia may occur, as in the clinical vignette. IgG positive wAIHA is more common, although up to 38% of cases may be DAT negative; anemia is usually very severe requiring transfusions, and prednisone 1.5 mg to 2 mg/kg per day along with checkpoint inhibitors (CPIs) discontinuation. Refractoriness and mortality rates remain high (both about 17%) [43,44]. 

### 5.5. Post-Transplant AIHA

*Clinical vignette 10-part 2*: Nine months after Evans syndrome recovery, the patient received haploidentical allogenic hematopoietic stem cell transplant (HSCT) from his brother with good engraftment and grade 2 acute skin graft versus host disease only. Nine months after transplant, he was admitted to the general ward due to very severe Evans relapse. Steroids, iv Ig, and rituximab were ineffective. At day 21 of admission, antibiotic prophylaxis was given, and laparoscopic emergency splenectomy was performed. Blood counts rapidly increased and the patient progressively recovered and was able to receive anti-capsulated vaccinations.

Another peculiar setting that may be observed is the development of autoimmune cytopenia after HSCT [45]. Factors favoring AIHA occurrence include the use of particular conditioning therapies preceding transplant, the subsequent immunosuppressive treatments, and the occurrence of HSCT complications such as viral infections reactivation. Mortality may be quite high and increases with frequently observed superimposed infections [48]. As in the clinical vignette, steroids are poorly effective (20% of cases only), whilst frontline rituximab gives better responses (up to 89%). Splenectomy may be considered in ultra-refractory cases, given the high surgical, infectious, and thrombotic risk [45,46,47]. Finally, it is worth reminding the passenger lymphocyte syndrome, which occurs after solid organ transplantation (kidney, liver, and heart-lung transplants). This condition is due to the lymphocytes present within the graft that are passively transferred to the recipient and produce antibodies against recipient RBCs. This condition is observed 3 to 24 days post-transplant and is usually mild and transient [15,46,49]. 

## 6. Conclusions

AIHA may represent a diagnostic and therapeutic challenge. In fact, beside typical presentation with anemia, hemolysis, and DAT positivity, many confounders (nutrients deficiencies, chronic liver or kidney diseases, infections, and cancers) may be present, making the differential diagnosis harder. If DAT is negative, other acquired (mechanic, infections, paroxysmal nocturnal hemoglobinuria (PNH), etc.) or congenital (membrane and enzyme defects) causes of hemolysis should be excluded, and diagnosis may be supported by the response to steroids. Secondary cases (drugs, infections, lymphoproliferative disorders, immunodeficiencies, etc.) should always be suspected and investigated with careful anamnesis and physical examination, followed by specific tests in selected cases, including autoantibody screening, Ig levels, ultrasound or CT studies, and bone marrow evaluation. The latter is also advised to evaluate dysplastic features or fibrosis that may have a therapeutic and prognostic importance, particularly in reticulocytopenic patients. AIHA may complicate specific clinical settings, including ICU admission, post-transplant, and treatment with novel anti-cancer drugs. These cases are often severe, more frequently DAT negative, and require multiple treatments in a short time. Finally, AIHA is usually mild and steroid-sensitive in the pregnant woman, although refractory cases have been described. With this in mind, the broadest diagnostic funnel will enable the correct identification and management of all AIHA difficult cases.

## Figures and Tables

**Figure 1 jcm-09-03858-f001:**
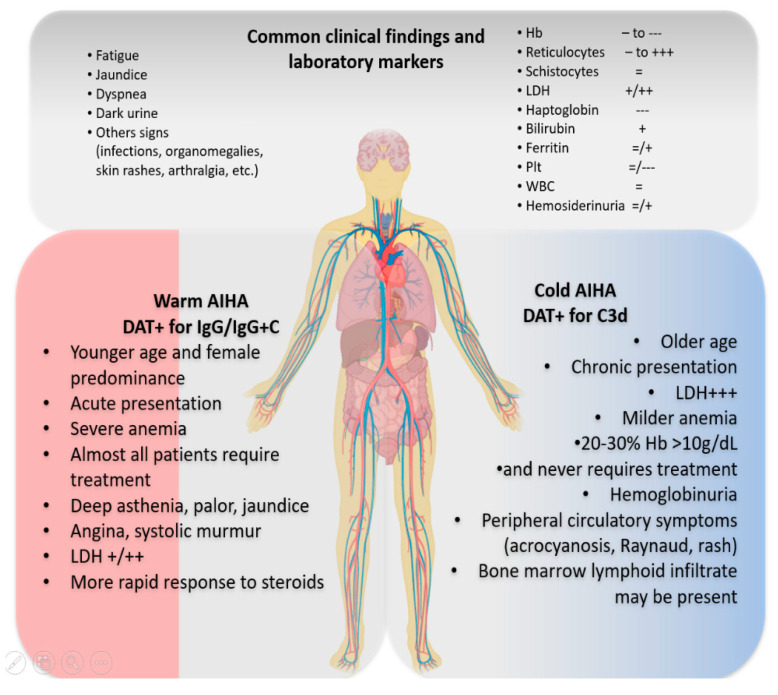
Clinical and laboratory findings in patients with primary autoimmune hemolytic anemia (AIHA). Symptoms and signs related to anemia and hemolysis are common to warm (wAIHA) and cold (cAIHA) forms. Hemoglobin (Hb), other blood counts (platelets Plt, white blood cells WBC), hemolytic markers (lactate dehydrogenase LDH, reticulocytes, haptoglobin, bilirubin, and hemosiderinuria), ferritin and schistocytes can be either normal =, slightly reduced -, reduced --, and very reduced --- or slightly increased +, increased ++, and very increased +++. In wAIHA, direct antiglobulin test (DAT) is usually positive for IgG with or without complement; cAIHA are DAT positive of complement fraction C3d. Some clinical and laboratory features are different and help to reach the diagnosis.

**Figure 2 jcm-09-03858-f002:**
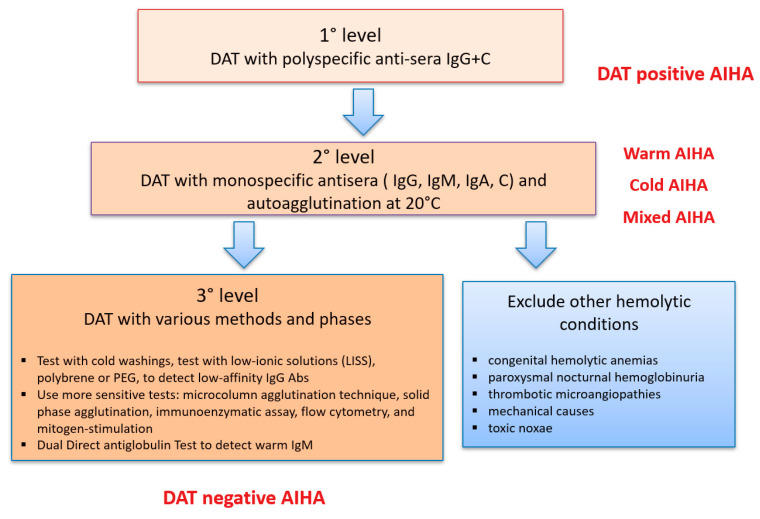
The various levels of direct antiglobulin test (DAT) study. The 1st level is DAT with polyspecific anti-sera that in the presence of hemolysis and anemia leads to the diagnosis of autoimmune hemolytic anemia (AIHA). The next and fundamental step (2nd level) is DAT with monospecific anti-sera and autoagglutination at 20 °C that allows the distinction of warm (positive for IgG+ or −C at low titer, autoagglutination negative), cold (positive for C and autoagglutination positive), and mixed forms (positive for IgG + C at high titer and autoagglutination positive). For negative cases, it is required to perform the DAT with various methods and phases (3rd level) and exclude all the other causes of hemolysis. Despite all the tests, up 5–10% of cases would remain DAT negative. Ig: immunoglobulins; PEG: polyethylene glycol; Abs: antibodies.

**Table 1 jcm-09-03858-t001:** Secondary conditions associated with autoimmune hemolytic anemia (AIHA).

Exogenous Predisposing Factors		
Infections	Frequency	Comments
Parvovirus B19; HCV; HAV; HBV; HIV Mycoplasma spp.; Tuberculosis; Babesiosis; Brucellosis; Syphilis; EBV; Respiratory Syncytial Virus	0.02% to 20%	ParvoB19 infection and HCV and its treatment correlate with AIHA development; case reports of association with AIHA are available for the other infectious agents.
Drugs		
Antibiotics (penicillins, cephalosporins, etc.), cytotoxic drugs (oxaliplatin, etc.), antidiabetics (metformin), anti-inflammatory drugs (diclofenac, etc.), neurologic drugs (α-methyldopa, L-dopa, chlorpromazine, etc.), cardiologic drugs (procainamide, etc.)	Case reports and reviews	Various mechanisms are demonstrated: hapten and drug absorption mechanisms; Immune/ternary complex mechanisms; autoantibody mechanism; non-immunologic protein formation; unknown mechanisms.
CLL therapy: fludarabine and Tyrosine kinase inhibitors	6–21%	Fludarabine induced AIHA may be avoided by rituximab association. Ibrutinib was associated to low risk of AIHA development in registrative trials in CLL
Vaccines		
Vaccines	0.8/100,000 person-years	AIHA was the rarest autoimmune complication in a population study
Endogenous predisposing factors		
Congenital Syndromes and Immunodeficiencies		
Kabuki syndrome and Hemoglobinopathies	4–6%	AIHA and ITP are the most frequent autoimmune complications of Kabuki Syndrome; DAT positivity is frequent, but clinically overt AIHA is rarer in thalassemia (particularly beta intermedia, alloimmunized and transfused pts)
ALPS; CVID; IgA deficiency	2–70%	AIHA is the most frequent autoimmune complication together with ITP and ES
Genes involved in PIDsTNFRSF6, CTLA4, STAT3, PIK3CD, CBL, ADAR1, LRBA, RAG1, and KRAS	40% of pediatric ES	Majority of pediatric ES display somatic mutations found in immunodeficiencies
Genetic Findings		
HLA I and II	Case series	HLA-B8 and BW6 are strongly associated to wAIHA.
IGHV and IGKV region	>60% cAIHA	Specific IGVH and IGKV regions are related to AIHA development
TCRG and TCRB	50%	Pathogenic T-cells are clonally restricted in AIHA
CTLA-4 exon 1	73%	CTLA-4 signaling is defective in AIHA, particularly in CLL cases
Cytokine polymorphisms	41%	AIHA shows higher frequency of LT-α (+252) AG phenotype
KMT2D and CARD11	69% and 31% of cAIHA tested	Autoreactive B-cells display somatic mutations favoring proliferation
Mixed (host and environmental) predisposing factors		
Autoimmune Diseases		
SLE, Systemic sclerosis; autoimmune thyroiditis; Sjogren Syndrome; IBDs; Autoimmune hepatitis/Primary biliary cirrhosis	1.4–14%	AIHA frequency is higher in pediatric than in adult patients with SLE. AIHA may be rarely associated to systemic sclerosis or Sjogren syndrome, Hashimoto thyroiditis and Graves’ disease, ulcerative colitis, and autoimmune hepatitis.
Lymphoproliferative Disorders	Frequency	Results
Chronic lymphoid leukemia and NHL	5–20%	Autoimmune cytopenias may frequently complicate chronic lymphoproliferative disorders and usually correlate with advanced disease and high biologic risk (unmutated IGHV status, stereotyped IGHV frames, and chromosome 17p and/or 11q deletions)
Solid Cancers		
Thymoma; Ovarian/Prostate	1.29–30% autoimmune phenomena	Thymoma and prostate and ovarian carcinomas have the highest association with autoimmunity

AIHA: autoimmune hemolytic anemia; wAIHA: warm AIHA; cAIHA: cold AIHA; ES: Evans syndrome; ITP: immune thrombocytopenia; DAT: direct antiglobulin test; CLL: chronic lymphocytic leukemia; ALPS: autoimmune lymphoproliferative syndrome; CVID: common variable immunodeficiency; SLE: systemic lupus erythematosus; IBDs: inflammatory bowel syndromes; HCV, HAV, HBV hepatitis C, A and B viruses; HIV: human immunodeficiency virus; EBV: Ebstein Barr virus.

**Table 2 jcm-09-03858-t002:** Autoimmune hemolytic anemia (AIHA) in peculiar clinical settings.

Setting	Clinical Presentation	Diagnostic Issues	Therapeutic Hints	Reference
AIHA in the intensive care unit 0.05% of primary AIHA cases	Very severe anemia Massive hemolysis Unresponsiveness to transfusion Multi-organ failure	Factors associated with severity are concomitant immune thrombocytopenia, severe infections, and thrombosis	Intensive support with transfusion, along with steroid boli, iv Ig, rituximab, erythropoietin, and plasma-exchange may be required	[11,12,26,27]
AIHA with reticulocytopenia 20% of AIHA cases	DAT positive AIHA with inadequate reticulocytosis typical of patients with severe or very severe AIHA	BMRI < 121 s [(absolute reticulocyte count) × (patient’ Hb/normal Hb)] Bone marrow mandatory to exclude underlying lymphoid or myeloid neoplasm MS-DAT may show anti-erythroblasts autoantibodies Endogenous EPO levels generally inadequate	Along with AIHA treatment recombinant EPO, i.e., alpha-epoetin 40,000 UI/week subcutaneously may be effective in up to 70% of cases	[13,28,29]
AIHA in pregnancy 1 in 50,000 pregnancies	Acute Hb drop during pregnancy or after delivery with alteration of hemolytic markers	Exclude HELLP and thrombotic microangiopathies DAT negative cases are also described	Typically responds to conventional treatment (steroids and iv Ig) and often resolves after delivery Evaluate thrombotic riskConsider that the newborn may be DAT+ in wAIHA since IgG cross the placenta	[30,31,32,33,34,35,36,37,38,39]
Novel anti-cancer drugs	Mono- or bilineage cytopenia	History of therapy with anti-PD1, anti-PDL1, or anti-CTLA4 MoAb Most of cases IgG wAIHA DAT may be negative in up to 38% of cases	Prednisone 1.5 mg to 2 mg/kg per day along with CPIs discontinuation is recommended	[40,41,42,43,44]
Post-transplant AIHA Passenger lymphocyte syndrome 9–70% of solid organ transplant	3 to 24 days post-transplant transient hemolysis	Mainly group O donors, though few cases have been described in AB recipients with non-AB donors risk of hemolysis ranging from 9 to 70% (kidney < liver < heart-lung transplants)	The syndrome is self-limiting and usually requires supportive treatment only	[15]
Hematopoietic stem cell transplant (HSCT) 2–4% of HSCTs	Severe immune hemolysis 3–10 months post-HSCT	Warm forms develop later (6–18 months) than cold forms (2–8 months)	Steroids are effective only in 20%, early rituximab is advised, case reports of novel drugs have been described	[45,46,47]

wAIHA: warm AIHA; cAIHA: cold AIHA; BMRI: bone marrow responsiveness index; DAT: direct antiglobulin test; MS-DAT: mitogen stimulated DAT; EPO: erythropoietin; MoAb: monoclonal antibody; HSCT, hematopoietic stem cell transplant; iv Ig: intravenous immunoglobulins; CPIs: checkpoint inhibitors; HELLP: syndrome of hemolysis, elevated liver enzymes, and low platelet count; anti-PD1: programmed cell death protein 1; PDL1: programmed death-ligand; anti-CTLA4: cytotoxic T-lymphocyte antigen 4.

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
