# Peer review of "Difficult Cases of Autoimmune Hemolytic Anemia: A Challenge for the Internal Medicine Specialist"

_jcm, 2020, doi:10.3390/jcm9123858_

Round 1

Reviewer 1 Report

This is a very nice review of autoimmune hemolytic anemia that appears to be targeted to the non-hematologist internal medicine specialist or possibly to hematology trainees. A strength of the manuscript is the use of illustrative clinical vignettes.

GENERAL:

1. It seems to me that there are some areas in the manuscript where the non-hematologist specialist could confuse alloimmune effects and autoimmune effects. The manuscript would benefit from a paragraph either in the Introduction or following the Introduction briefly discussing the differences between autoimmune hemolysis alloimmune hemolysis and how they are distinguished.

2. There are some minor issues of English usage. For example, there are some long sentences with several clauses separated by commas when at least some of these should be separated by semicolons, and some minor issues of verb tense disagreement and errors of idiom. These would likely be caught with more careful proofreading.

SPECIFIC:

1. Vignette 1:
-When laboratory values are stated, the normal range for those values should be stated.
-Which "iron levels" were normal?
-Which "vitamins" were normal?
-"Hemorrhagia" is not a standard English term. "Bleeding" or "Hemorrhage" should be used instead.

2. Vignette 2:

- It might be more accurate to say that splenectomy is "ineffective" than to say it is "contraindicated". " Contraindicated" implies a specific negative outcome other than just lack of efficacy.

3. Vignette 3:
- Very few readers (at least in the US) will know what "ex juvantibus" means. It would be more clear to either say "a therapeutic trial" or "a trial of empiric therapy".
-This is one of the places where confusion between alloimmune and autoimmune effects might occur to the nonhematologic reader. These are not exactly "false positive" DAT results so much as they are "artifactually positive" DAT results.

4. Vignette 4:
-The current terminology for the syndrome of concurrent autoimmune hemolytic anemia and immune thrombocytopenia is "Evans" syndrome without the use of an apostrophe.

5 Vignette 8:
-" ...reduced bone marrow stemness" is not the correct English terminology for this concept. It should be referred to as "reduced bone marrow reserve" or "reduced bone marrow stem cell reserve".

Author Response

We would like to thank the Referee for the positive feedback and for the careful revision. As suggested, we checked and corrected the mistakes throughout the manuscript.

GENERAL:

  1. It seems to me that there are some areas in the manuscript where the non-hematologist specialist could confuse alloimmune effects and autoimmune effects. The manuscript would benefit from a paragraph either in the Introduction or following the Introduction briefly discussing the differences between autoimmune hemolysis alloimmune hemolysis and how they are distinguished.

As suggested, we implemented the distinction between auto and allo-immunity within paragraph 2 and 3.

“Importantly, autoantibodies directed against autologous erythrocytes and detected by DAT, have to be distinguished from alloantibodies that may be found in patient serum and react with donor erythrocytes in the “indirect anti-globulin test” (IAT). The latter may be positive also in a fraction of AIHA patients as described later.”

“Finally, the above mentioned indirect antiglobulin test is important to detect autoantibodies in patient’s serum, and also to reveal the presence of alloantibodies (reported in 1/3 of AIHA and possibly causing transfusion reactions) [9, 13]. The detection of a positive IAT requires further studies to identify their specificity and confirm the auto- or alloreactive nature; moreover, these studies will allow to find matched packed red cells units to either avoid transfusion reactions and further immunization.”

  1. There are some minor issues of English usage. For example, there are some long sentences with several clauses separated by commas when at least some of these should be separated by semicolons, and some minor issues of verb tense disagreement and errors of idiom. These would likely be caught with more careful proofreading.

We thank the Referee for the thorough revision and corrected the mistakes and the various sentences as suggested. Moreover, we specified iron and vitamin tests to be made as well as the normal ranges of cited laboratory values.

SPECIFIC:

  1. Vignette 1:

-When laboratory values are stated, the normal range for those values should be stated.

-Which "iron levels" were normal?

-Which "vitamins" were normal?

-"Hemorrhagia" is not a standard English term. "Bleeding" or "Hemorrhage" should be used instead.

  1. Vignette 2:

- It might be more accurate to say that splenectomy is "ineffective" than to say it is "contraindicated". " Contraindicated" implies a specific negative outcome other than just lack of efficacy.

We substituted the term “contraindicated” with “ineffective”.

  1. Vignette 3:

- Very few readers (at least in the US) will know what "ex juvantibus" means. It would be more clear to either say "a therapeutic trial" or "a trial of empiric therapy".

-This is one of the places where confusion between alloimmune and autoimmune effects might occur to the nonhematologic reader. These are not exactly "false positive" DAT results so much as they are "artifactually positive" DAT results.

We agree with the Referee and clarified the sentences as suggested.

  1. Vignette 4:

-The current terminology for the syndrome of concurrent autoimmune hemolytic anemia and immune thrombocytopenia is "Evans" syndrome without the use of an apostrophe.

Corrected as suggested.

5 Vignette 8:

-" ...reduced bone marrow stemness" is not the correct English terminology for this concept. It should be referred to as "reduced bone marrow reserve" or "reduced bone marrow stem cell reserve".

Corrected as suggested.

Reviewer 2 Report

This is a comprehensive and well written account of the clinical challenges faced when diagnosing and treating patients with AIHA.

Some minor changes are required.

p2: Define the abbreviation of ULN (upper limit of normal)

p3: 'referred from the general practitioner' should read 'referred from a general practitioner'

p5: ellyptocytes 'elliptocytes'.    Line 10:  'if the clinical suspect' should read 'if the clinical suspicion'

Table 1: Tyrosin should read tyrosine

p9: 'hemolyitic' should read 'hemolytic'

p10: Line 2 - delete 'to' before nutrient deficiency

Author Response

We thank the Referee for the revision, for the positive feedback and for the helpful suggestions. As advised, we corrected misspells and clarified acronyms throughout the text.
